# Recent Developments and Current Status of Commercial Production of Fuel Ethanol

**Tuan-Dung Hoang [1],[*] and Nhuan Nghiem [2]**

[1] Department of Environmental Security and Technology, Hanoi School of Business and Management, Vietnam National University, 144-Xuan Thuy Street, Hanoi 10000, Vietnam

[2] Biosystems Engineering Program, Department of Environmental Engineering and Earth Sciences, Clemson University, Clemson, SC 29631, USA; nnghiem@g.clemson.edu

[*] Correspondence: tuandung@vnu.edu.vn or tuandunghoang@gmail.com; Tel.: +84-936414637

**Abstract:** Ethanol produced from various biobased sources (bioethanol) has been gaining high attention lately due to its potential to cut down net emissions of carbon dioxide while reducing burgeoning world dependence on fossil fuels. Global ethanol production has increased more than six-fold from 18 billion liters at the turn of the century to 110 billion liters in 2019, only to fall to 98.6 billion liters in 2020 due to the pandemic. Sugar cane and corn have been used as the major feedstocks for ethanol production. Lignocellulosic biomass has recently been considered as another potential feedstock due to its non-food competing status and its availability in very large quantities. This paper reviews recent developments and current status of commercial production of ethanol across the world with a focus on the technological aspects. The review includes the ethanol production processes used for each type of feedstock, both currently practiced at commercial scale and still under developments, and current production trends in various regions and countries in the world.

**Keywords:** fuel ethanol; renewable energy; biobased feedstocks; lignocellulosic biomass; fermentation process; processing options; commercialization; production status; climate change; environmental security

## 1. Introduction

Ethanol ($C_2H_6O$) is a simple liquid alcohol that is formed from the fermentation of sugars in their natural occurrences or being derived from starch-rich grains or lignocellulosic feedstocks. Ethanol is also called ethyl alcohol, grain alcohol, or simply alcohol, and is used as a disinfectant, an organic solvent, a chemical feedstock, and a transportation liquid fuel. Ethanol is produced in various countries in the world and its global quantity has changed from 110 billion liters in 2019 to 98.6 billion liters in 2020 due to the pandemic [1,2]. When ethanol is blended with 95% gasoline it can reduce about 90% $CO_2$ and 60–80% $SO_2$ [3]. This helps the world to solve some of the problems of air pollution, reduces the levels of greenhouse gases that are causing climate change and maintains environmental security. Currently, ethanol is produced commercially from a variety of feedstocks via fermentation where the yeast *Saccharomyces cerevisiae* is utilized to ferment the sugars derived from the starch in corn and other grains or those that already are present in sugar cane and sugar beets [4].

Crude oil and natural gas are traditionally the main raw materials for producing fuels and industrial chemicals. However, human dependency on fossil fuels has become a critical issue worldwide and it is essential to explore new sustainable and alternative sources to solve the world's environmental problems [5]. Ethanol is an important source of biofuels and the ethanol development and implementation is within the scope of circular bioeconomy since biofuels can substitute for fossil carbon by bio-based carbon in biomass from agriculture, forestry and municipal wastes. In terms of sustainability of bioethanol production compared to fossil-based fuels, ethanol development is a complementary

approach and based on improved resources with higher eco-efficiency and reduced GHGs footprint [6]. Furthermore, the increase of liquid fuels supplying the energy demand with regards to economic and environmental concerns make ethanol a good energy alternative for many countries.

There are commonly three groups of materials that can be used for manufacturing ethanol, which are: (a) feedstocks which contain substantial amounts of readily fermentable sugars, (b) starches and fructans, and (c) cellulosic materials. However, in-depth discussions on the different feedstocks vis-à-vis production technologies and current development of ethanol worldwide are still lacking. The main objective of this review is to investigate the recent development of ethanol production in countries with significant levels of ethanol outputs. The review also examines the current technologies for ethanol production and those being developed toward commercial implementation.

## 2. Production Technologies

Three types of feedstock can be used for ethanol production. These include sugar-based feedstocks (sugar cane, sugar beet), starched-based feedstocks (corn, barley, wheat, other grains), and lignocellulosic feedstocks (agricultural residues, forest residues, dedicated energy crops, municipal solid wastes). The ethanol produced from sugar-based and starch-based feedstocks is referred to as first-generation ethanol whereas the ethanol produced from biomass feedstocks is referred to as second-generation ethanol.

### 2.1. Sugar-Based Feedstocks

The juice extracted from sugar cane has been used for ethanol production. Part of the juice is used for sugar manufacture in an adjacent sugar manufacturing plant and the remaining is used for ethanol production such as the case of Brazil, which is the second largest ethanol producer in the world. Juice extraction can be practiced through either crushing (roll mills) or diffusion (diffuser). The decision to produce sugar and/or ethanol is made by individual plants from harvest to harvest. This is a very important issue since once the producing units adjust their plants to produce a set ratio of sugar/ethanol in a given year, it is very difficult to change during the harvest season [7]. Bagasse, which is the solid residue obtained after juice extraction, has been combusted to generate energy for internal uses, but also has the potential as a lignocellulosic feedstock for additional ethanol production. Molasses, which is a by-product of sugar cane processing, can also be used for ethanol production.

Typically, sugarcane contains 12–17% total sugars on a wet weight basis with 68–72% moisture. About 90% of the sugars are sucrose with glucose and fructose making up the balance. All these three sugars are readily fermentable by the yeast *S. cerevisiae* to produce ethanol. During the fermentation process, the yeast produces the enzyme invertase and uses it to convert sucrose to glucose and fructose. Ethanol production by *S. cerevisiae* is carried out via the glycolytic pathway (also known as the Embden-Meyerhof-Parnas or EMP pathway). In the simplest form, production of ethanol from glucose can be expressed by the following equation:

$$C_6H_{12}O_6 + 2\,P_i + 2\,ADP \rightarrow 2\,C_2H_5OH + 2\,CO_2 + 2\,ATP + 2\,H_2O$$

$$glucose \rightarrow 2\ ethanol + 2\ carbon\ dioxide + energy$$

According to the above equation, the maximum theoretical yield of ethanol produced from 1 g of glucose is 0.511 g ethanol. In practice, it is observed that the actual yield is always lower since not all of the glucose consumed is converted to ethanol [8].

The fermentation process can be either batch/fedbatch or continuous. The Melle-Boinot batch process, which was developed in the 1930′s, is most commonly used. Its main characteristic is total yeast recycle, normally by centrifugation. The very high yeast densities (10–14% $w/v$), which are obtained by yeast recycle, allow reduced growth, high ethanol yield and very short fermentation time (6–10 h). The continuous process with cell

recycle was developed in the 1980′s to replace the batch process [9]. The majority of the commercial plants in Brazil, however, still employ the batch/fed-batch process. The ethanol produced is recovered by distillation followed by molecular sieve or other dehydration technology to produce anhydrous ethanol (>99.5%). The stream coming out at the bottom of the distillation columns is called vinasse. Vinasse normally is spread on the sugar cane fields for use as irrigation water and fertilizer. However, this practice may cause extensive pollution of the environment due to the high organic contents, dark color, dissolved solids and other compounds that are toxic under certain conditions. Methods for treatment of vinasse prior to their disposal have been developed [10].

Sugar beet is grown in regions with temperate climates and is one of the main feedstocks used for ethanol production in Europe. Sugar beet has a conical, white and fleshy root, which serves as a reservoir for sugar. The root contains about 75% water and between 15% and 21% total sugars (wet weight basis). Due to the similarities in chemical compositions of sugar solutions extracted from sugar cane and sugar beet, those obtained from sugar beet also can be fermented to ethanol by industrial yeast at high efficiencies. The fermentation processes are very similar in both cases. Similar to sugar cane, ethanol production from sugar beet can be integrated with sugar production.

### 2.2. Starch-Based Feedstocks

Corn is used almost exclusively for ethanol production in the United States where ethanol is produced from corn by either the *wet-milling* or *dry-grind* process. The United States is the largest producer of ethanol in the world where the majority of the commercial ethanol plants use the dry-grind process. Ethanol can also be produced from other grains by the dry-grind process.

In the dry-grind process, after grinding, water and a thermostable $\alpha$-amylase are added to the ground corn. In the next step, which is called pre-liquefaction, the slurry then is brought up to 60–70 °C (warm cook) or 80–90 °C (hot cook). The slurry, which is referred to as the corn mash, is held at these temperatures for about 30–45 min. The swelling and hydration of the starch granules cause dramatic increase of the slurry viscosity and loss of crystallinity of the granule structures. In the next step, which is called liquefaction, the mash is maintained at 85–95 °C for a period of time or forced through a continuous jet cooker at 140–150 °C. At the end of the liquefaction, starch is hydrolyzed to short-chain dextrins (two to four glucose units). The temperature of the mash is lowered to 32 °C and the pH adjusted to approximately 4.5. The mash then is placed in a fermentor. Glucoamylase and the yeast culture from the yeast propagation tank are also added. Urea may be added as a nitrogen source. The process combining enzymatic hydrolysis and fermentation is called simultaneous saccharification and fermentation (SSF). The SSF is a batch process which typically is run for about 50–60 h. The final ethanol concentration is about 15 % (*v/v*). Commonly used commercial glucoamylase formulations also contain proteases. These enzymes break down organic nitrogen sources in the corn slurry and release additional nutrients, which help to improve fermentation efficiency and ethanol yield. The ethanol produced is recovered by distillation followed by molecular sieve as described previously. The bottom stream from the distillation columns is called stillage. This stream is dried to a moisture content of about 10% to produce distillers dried grains with solubles (DDGS), which are sold or used as animal feed. DDGS is the most important co-product in a dry-grind ethanol plant. The other co-products are the carbon dioxide produced in the fermentation process and corn oil. The dry-grind process is shown in Figure 1.

In the wet-milling process, corn grains are soaked in water containing 0.1 to 0.2% $SO_2$ at 52 °C (125 °F) for 24 to 40 h. After this so-called steeping step, the softened grains are then ground gently to break up the kernels. The less dense germs, which contain about 40 to 50% oil, are recovered in a hydrocyclone system and processed into value-added co-products. After the germs are removed, the slurry was subjected to further grinding to loosen the starch and gluten from the remaining fiber. The slurry is screened to remove

the fiber, which then is washed and pressed to about 60% moisture. The water from the steeping step, which is called light steep water (LSW), is concentrated in evaporators to produce heavy steep water (HSW). The HSW is dried together with the recovered fiber and the resultant product and is sold to the livestock industry as *corn gluten feed*, which typically contains about 21% proteins. After fiber removal, the gluten, which is lighter, is separated from the starch in a centrifuge. The recovered gluten is concentrated from, cooled to 35 °C and filtered on a rotary vacuum filter to produce a gluten cake of 60% moisture. The cake is dried to 10% moisture to produce *corn gluten meal*. This feed product contains at least 60% protein and 1% fat, and up to 3% fiber. It is used in poultry feed because of its high protein and xanthophyll and low fiber contents.

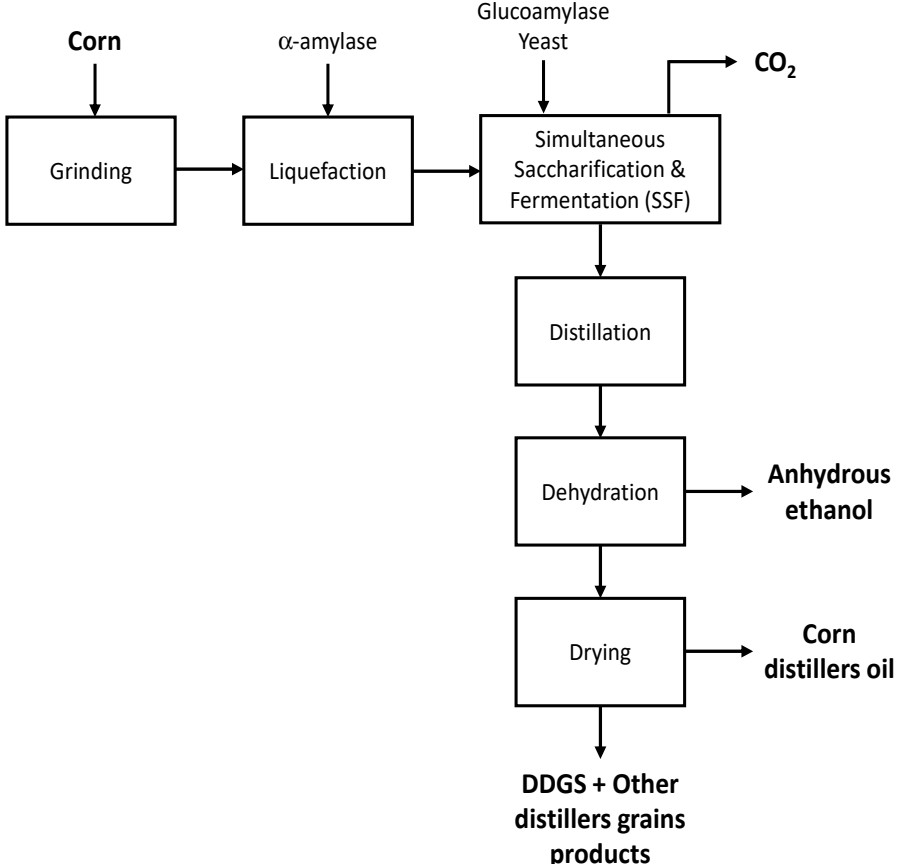

**Figure 1.** The dry-grind process for ethanol production from corn.

The starch is washed and processed through a series of up to 14 hydrocyclones to remove impurities. The final product, which is 99.5% pure starch, then is used for ethanol fermentation in dedicated plants or processed further to produce modified starch, corn syrup (CS) and high fructose corn syrup (HFCS) in integrated plants. The wet-milling process is shown in Figure 2.

The hydrolysis of starch to produce glucose can be expressed by the following equation:

$$(C_6H_{10}O_5)_n + n\,H_2O \rightarrow n\,C_6H_{12}O_6$$

where n is the number of glucose residues in the starch molecule.

In a starch molecule, n is a very large number. The theoretical yield or conversion factor for glucose is 1.11 g glucose/g starch. In addition, the theoretical yield for ethanol is 0.57 g ethanol/g starch.

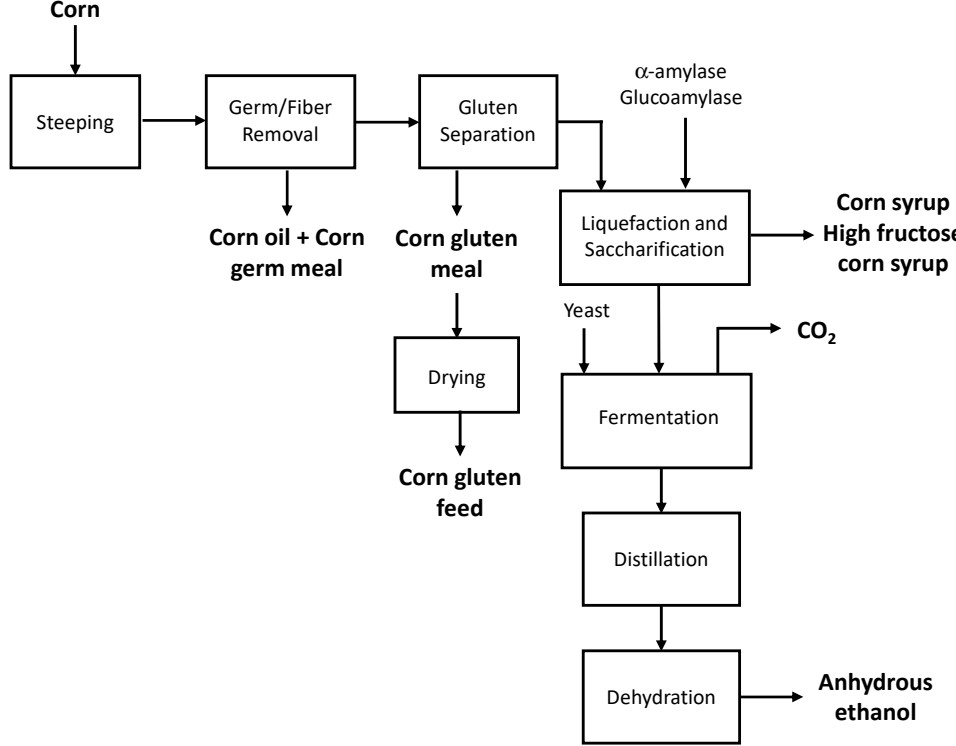

**Figure 2.** The wet-milling process for ethanol production from corn.

In the United States, it is commonly accepted that one bushel of corn (56 lbs or 25.4 kg) of about 15% moisture processed by the dry-grind process will produce 2.9 gallons (11.0 L) denatured ethanol, 15.2 lbs (6.9 kg) distillers grains animal feed (10% moisture), 0.8 lbs (0.36 kg) distillers corn oil, and 1.1 lbs (0.5 kg) captured carbon dioxide [11].

Many efforts have been made to improve ethanol production with the focus on the dry-grind process. These improvements are discussed briefly below. Syngenta developed the Enogen® corn hybrids containing α-amylase, which helps to reduce the viscosity of the corn mash and reduce/eliminate the need of externally added α-amylase [12].

- Lallemand, in collaboration with other companies, developed the TransFerm® Yield+ *S. cerevisiae*, which produces the glucoamylase needed for starch hydrolysis and improves ethanol yield up to 4% as a result of 30% reduction in glycerol synthesis. The reduced glycerol levels are not expected to have an adverse effect on the yeast tolerance of osmotic stress caused by ethanol [13]. The newer strain, TransFerm CV5, is a genetically modified yeast strain that produces high levels of glucoamylase and trehalase expression, which can meet between 80% and 100% of the enzymes required in fermentation [14].
- Novozymes developed the glucoamylase-producing Innova Drive *S. cerevisiae*, which could tolerate 98 °F (36.7 °C), 37% solid loading, and 6 g/L organic acids [15]. Other strains of the Innova product series also have been developed with improved tolerance of heat (up to 40 °C) and resistance to high levels of solids and high concentrations of glucose, ethanol and organic acids [16].

Several companies, which include Poet (formerly Broin), Cereal Process Technologies (CPT), Buhler, Renessen, and FWS Technologies, have developed dry fractionation processes. All dry corn fractionation processes are fundamentally similar. Prior to the conventional dry-grind process, corn is ground and subsequently fractionated into fermentable and non-fermentable fractions. Only the fermentable fraction is used in the dry-grind process for ethanol production. The non-fermentable fractions are separated by various methods and recovered as bran and germ co-products, which are sold for use in animal feeds. The DDGS obtained after ethanol recovery has higher protein, but lower oil and

fiber contents compared to the product obtained in a traditional dry-grind process without corn fractionation [8].

- D3MAX developed a process for conversion of corn fiber and the unconverted starch in the stillage to additional ethanol. The D3MAX process is a "bolt-on" process, which means it can be added to an existing dry-grind ethanol plant. The key step of this process is the pretreatment of the wet cake obtained by centrifugation of the stillage with dilute acid. The pretreated wet cake then is subjected to enzymatic hydrolysis and fermentation, which uses a genetically modified yeast capable of utilizing both glucose and xylose for ethanol production [17].
- Quad County Corn processors (QCCP) developed the Cellerate process, which is also a bolt-on process. This process is very similar to the D3MAX process, except that the whole stillage rather than the wet cake is pretreated with dilute acid prior to enzymatic hydrolysis and fermentation. It was reported that when Enogen® corn is used (in collaboration with Syngenta), compared to the traditional dry-grind process using regular corn, the Cellerate process resulted in 6% increase in ethanol yield, 15% increase in throughput, and 20% reduction in energy consumption, and produced 1.6 lbs (0.73 kg) corn oil per bushel plus a DDGS with higher protein and lower fiber [18].
- Edeniq developed the Intellulose® process using their proprietary enzyme mixtures to produce ethanol from the previously unconverted starch and fibers. It was reported that the technology resulted in a 2–4.5% increase in ethanol production. Edeniq also developed an analytical technique to directly measure ethanol production from the lignocellulosic fractions in the corn kernel [19,20].

### 2.3. Lignocellulosic Feedstocks

Lignocellulosic feedstocks are also referred to as lignocellulosic biomass or simply biomass. In this review, the term biomass is used. Biomass consists of three main components, which are cellulose, hemicellulose, and lignin. The compositions of representative biomass feedstocks and the theoretical ethanol yield by the biochemical conversion are summarized in Table 1.

**Table 1.** Compositions (dry basis) of representative biomass feedstocks and theoretical ethanol yield by the biochemical conversion [8].

| Lignocellulosic Feedstocks | Cellulose (%) | Hemicellulose (%) | Lignin (%) | Theoretical Ethanol Yield (L/MT) |
|---|---|---|---|---|
| **Forest products** | | | | |
| Hardwood | 46.2 | 29.2 | 22.0 | 546 |
| Softwood | 41.2 | 26.8 | 29.8 | 493 |
| **Woody energy crops** | | | | |
| Willow | 42.5 | 22.0 | 26.0 | 467 |
| Eucalyptus | 54.1 | 18.4 | 21.5 | 524 |
| Poplar | 52.1 | 27.5 | 15.9 | 576 |
| Pine | 46.0 | 25.5 | 20.0 | 518 |
| **Agricultural residues** | | | | |
| Corn stover | 35.2 | 25.1 | 23.7 | 437 |
| Rice straw | 43.4 | 27.9 | 17.2 | 517 |
| Barley straw | 41.0 | 26.6 | 21.3 | 490 |
| Wheat straw | 37.0 | 26.5 | 14.0 | 460 |
| Sugarcane bagasse | 41.6 | 25.1 | 20.3 | 483 |

There are two options for the conversion of biomass to ethanol. In the first option, cellulose and hemicellulose are hydrolyzed to fermentable sugars, which are subsequently fermented to produce ethanol. In the second option, the biomass is taken through a process called gasification. In this process, the biomass is heated with no oxygen or with oxygen significantly below the normally required for complete combustion. The product is a gas

which is mostly CO and $H_2$. The gaseous product is called synthesis gas or syngas, which can be converted to ethanol via either fermentation or chemical catalysis processes. The first option normally is referred to as the sugar platform and the second as the syngas platform.

In the sugar platform, only the carbohydrate components of biomass are used for ethanol production, whereas lignin is considered as a waste and normally is burned to generate energy. In the syngas platform, all three components are used for ethanol production. The key steps of the sugar platform and the syngas platform are shown in Figures 3 and 4, respectively.

Hydrolysis of cellulose and hemicellulose to generate fermentable sugars for ethanol fermentation can be performed with either chemicals or enzymes. The chemicals used for the hydrolysis of cellulose and hemicellulose include concentrated (>70%) $H_2SO_4$, which is used in the Arkenol process [21] and the Biosulfurol process [22], supercritical fluid, which is used in the Renmatix process [23], and γ-valerolactone (GVL), which is used in the GlucanBio process [24].

When concentrated $H_2SO_4$ was used for hydrolysis, the sugar product was a mixture of glucose and xylose whereas in the other two cases, two separate streams that contained mostly glucose and xylose, respectively, were formed. The fractionation of the two carbohydrate fractions would allow the use of the C6 sugar stream for ethanol production by the native *S. ceresiae* strains and the C5 sugar stream for production of higher-value co-products such as xylitol and astaxanthin [25].

Due to the rigid structure of biomass, which impedes enzymatic attacks, a process called pretreatment is required before the cellulose and hemicellulose components of biomass can be effectively hydrolyzed by enzymes. In the pretreatment process the rigid structure of biomass is opened up and amorphous regions are created. The results of the pretreatment process are improvements of the rates of the subsequent enzymatic hydrolysis and increases of the yields of the fermentable sugars. Various chemicals and reagents have been used for biomass pretreatment process research and development. Some of these chemicals and reagents together with their corresponding substrates are summarized in Table 2.

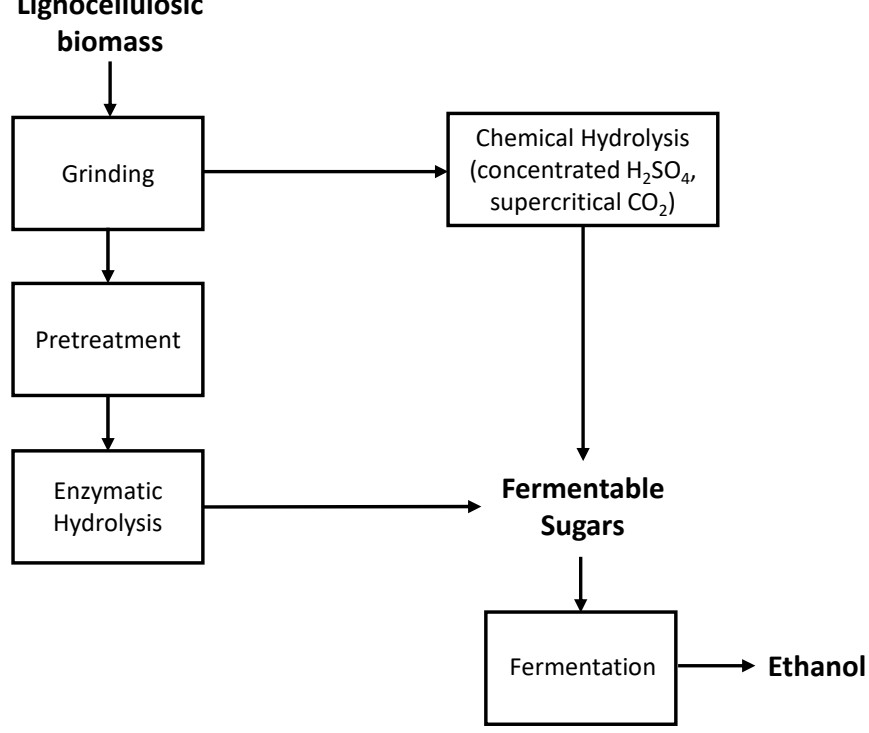

**Figure 3.** The sugar platform for ethanol production from biomass.

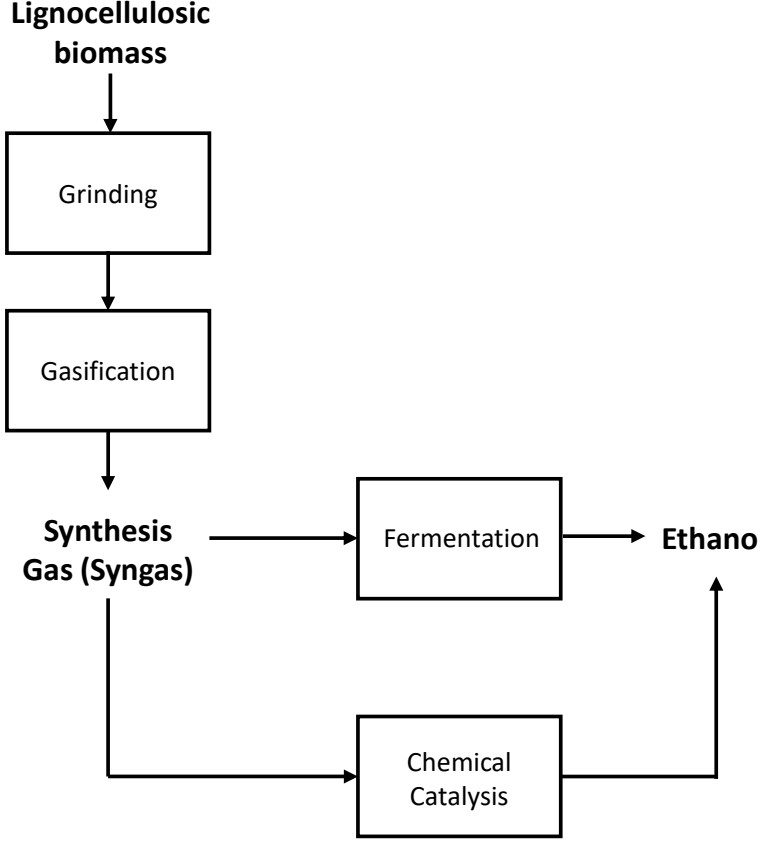

**Figure 4.** The syngas platform for ethanol production from biomass.

Among the many pretreatment processes there has not been a clear winner. The use of superheated steam, however, seemed to be a favorite. It has been used in the commercial Inbicon process for pretreatment of wheat straw [46]) and more recently in the Clariant cellulosic ethanol plant in Podari, Romania [47]. The pretreatment processes have been reviewed thoroughly in several reviews [8,48–50]. The fermentable sugars obtained in enzymatic hydrolysis of cellulose and hemicellulose consists mostly of glucose and xylose, plus arabinose, galactose, and mannose in significantly smaller amounts. Since the yeast *S. cerevisiae* cannot metabolize five-carbon sugars such as xylose, genetically engineered strains capable of utilizing both glucose and xylose for ethanol production have been developed for commercial applications [51].

Various types of commercial gasifiers are suitable for biomass gasification. The impurities and inhibitory compounds in the syngas produced have to be removed before the syngas can be used for ethanol production in either the microbial or catalytic process. Biomass gasification has been reviewed in detail by Ciliberti et al. [52]. The microorganisms that have been considered as most promising candidates for commercial syngas ethanol fermentation include *Clostridium ljungdahlii*, *Clostridium ragsdalei*, *Clostridium carboxydivorans*, *Clostridium coskatii*, *Clostridium autoethanogenum*, and *Alkalibaculum bacchi*. Some of these species only produce ethanol and acetic acid, whereas others can produce additional products, which include butyric acid, butanol, and 2, 3-butanediol as mentioned previously.

One of the key issues of ethanol production from biomass via fermentation, either in the sugar platform or the syngas platform, is the low achievable final ethanol concentration. Since it is most likely that the ethanol produced will be recovered by distillation, which is a very energy-intensive process, the minimum acceptable ethanol concentration in the input to the distillation unit is 50 g/L [8].

**Table 2.** Summary of biomass pretreatment chemicals and agents [26–45].

| Chemicals/Reagents | Biomass Type | References |
|---|---|---|
| **Dilute acids** | | |
| $H_2SO_4$ | Doulas fir chip | Nguyen et al. 1998 [26] |
| HCl | Sorghum straw | Herrera et al. 2003 [27] |
| $H_3PO_4$ | Sugarcane bagasse | Geddes et al. 2010 [28] |
| $HNO_3$ | Corn stover | Zhang et al. 2011 [29] |
| **Alkalines** | | |
| NaOH | Rice straw | Castro et al. 2017 [30] |
| KOH | Hybrid poplar | Gupta and Lee 2010 [31] |
| $NH_4OH$ | Corn stover | Le et al. 2021 [32] |
| Anhydrous $NH_3$ | Hybrid poplar | Balan et al. 2009 [33] |
| $Ca(OH)_2$ (Lime) | Corn stover | Kaar and Holtzapple 2000 [34] |
| **Basic Salts** | | |
| Green liquor | Sugarcane bagasse | Zhou et al. 2016 [35] |
| $Na_2CO_3/Na_2S$ | Sweet sorghum bagasse | Pham et al. 2018 [36] |
| $Na_2CO_3NaOH$ | Sweet sorghum bagasse | Nghiem and Toht 2020 [37] |
| $Na_3PO_4$ | Corn stover | Qing et al. 2016 [38] |
| **Water and Steam** | Spruce wood chips | |
| High pressure team | | Pielhop et al. 2016 [39] |
| Liquid hot water | Poplar | Li et al. 2017 [40] |
| **Organic solvents** | | |
| Methanol | Corn stover | Qing et al. 2017 [41] |
| Ethanol (with/without acid catalyst) | Various | Zhang et al. 2016 [42] |
| Ethylen glycol | Palm tree residues | Ariols et al. 2009 [43] |
| Glycerol | Rice straw | Trinh et al. 2016 [44] |
| $\gamma$-velerolactone | Hard wood | Shuai et al. 2016 [45] |

## 3. Global Ethanol Production

It is estimated that currently 60% ethanol is produced from corn, 25% from sugar cane, 3% from wheat, 2% from molasses, and the rest from other grains, cassava and sugar beets. The annual world fuel ethanol production in the last three years together with the production by the individual countries that contributed to at least 1% of the total production are summarized in Table 3. As indicated in Table 3, the top five ethanol producers before the COVID-19 pandemic were the United States, Brazil, the European Union (EU), China, and Canada. In 2020, India produced slightly more ethanol than Canada but the ranking of the top four ethanol producers did not change.

**Table 3.** Annual world fuel ethanol production (Mil. L) (Source: Renewable Fuels Association analysis of public and private data sources [53].

| Region | 2018 | 2019 | 2020 | % of 2020 World Production |
|---|---|---|---|---|
| United States | 16,091 | 15,778 | 13,926 | 53% |
| Brazil | 7990 | 8590 | 7930 | 30% |
| European Union (EU) | 1450 | 1370 | 1250 | 5% |
| China | 770 | 1000 | 880 | 3% |
| Canada | 460 | 520 | 428 | 2% |
| India | 430 | 510 | 515 | 2% |
| Thailand | 390 | 430 | 400 | 2% |
| Argentina | 290 | 280 | 230 | 1% |
| Rest of the world | 529 | 522 | 500 | 2% |
| Total | 28,400 | 29,000 | 26,059 | |

Note: The EU in this review refers to the union of 28 members, i.e., before the exit of the United Kingdom.

The number of production plants, plant type, feedstocks, and co-products of the ethanol industry in the top five ethanol-producing countries/regions are summarized in Table 4. In this table, all the data are for 2020. The case of the United States needs an explanation. Several corn dry-grind ethanol plants in the United States used a bolt-on

process such as the D3MAX or QCCP process discussed previously to convert the corn fibers, i.e., a biomass feedstock, and the non-converted corn starch to additional ethanol. The quantities of ethanol produced from each feedstock in these processes could not be determined separately. Therefore, all the ethanol produced in these plants are listed under corn ethanol production. In addition, some plants occasionally used small amounts of sorghum grains to mix with corn. The quantities of sorghum grains used were not known. Therefore, all the starch-based feedstock is listed under corn.

**Table 4.** The number of ethanol plants, plant type, feedstocks, and co-products in the top five ethanol-producing countries/regions [54–58].

| Country/Region | USA | Brazil | EU | China | India |
|---|---|---|---|---|---|
| Number of 1st-generation plants | 208 | 360 | 57 | 18 | 220 |
| Nameplate capacity (Bil L) | 65.8 | 42.8 | 8.15 | 6.58 | 3.50 |
| Capacity used (%) | 80 | 67 | 58 | 49 | 85 |
| Number of 2nd-generation plants | 3 | 3 | 3 | 1 | 0 |
| Nameplate capacity (Bil L) | 0.21 | 0.13 | 0.09 | 0.07 | |
| Capacity used (%) | n/a | 25 | 28 | 0 | |
| Feedstocks (1000 MT) | | | | | |
| Corn | 123,465 | 5995 | 6350 | 7100 | 0 |
| Other grains | 0 | 0 | 4300 | 900 | 0 |
| Cassava | 0 | 0 | 0 | 1000 | 0 |
| Sugarcane | 0 | 326,630 | 0 | 0 | 0 |
| Sugar beets | 0 | 0 | 7450 | 0 | 0 |
| Molasses | 0 | 0 | 0 | 0 | 6407 |
| Biomass | 0 | 0.178 | 200 | 200 | 0 |
| Co-products (1000 MT) | | | | | |
| Bagasse | 0 | 120,077 | 0 | 0 | 118,374 |
| Distillers grains | 29,437 | 1876 | 3332 | 2348 | 0 |
| Corn gluten feed | 3087 | 0 | 0 | 0 | 0 |
| Corn gluten meal | 605 | 0 | 0 | 0 | 0 |
| Corn Oil | 27.9 | 108.0 | 184 | 0 | 0 |

Note: n/a: Not available.

Ethanol production from the individual nations with significant annual ethanol outputs are reviewed in the following sections.

*3.1. North America*

3.1.1. The United States

In the United States, ethanol produced from corn reached 6.5 billion gallons (24.6 billion L) in 2007 and in 2012 this total doubled to approximately 13.2 billion gallons (50.0 billion L). In 2018 the United States exported a total of 6.5 billion L of ethanol, expanding the share of world exports of ethanol to 61%. Most ethanol produced in the United States is from starch-based crops by dry or wet-mill processing technologies. In 2020, there were 208 ethanol production plants in the United States with a total installed capacity of 17.44 billion gallons per year (66.0 billion L per year). The total production in 2020 was 13.8 billion gallons (52.2 billion L), which represented a 12.7% decrease from the production total of 15.8 billion gallons (59.8 billion L) in the previous year [11,59]. The decrease in ethanol production was mainly caused by the reduction in transportation fuel demand, which was the direct consequence of the COVID-19 pandemic. Whereas a number of plants were shut down, some plants modified the production process to produce an ethanol product which is suitable for use in hand sanitizers to cope with the economic downturn. It is expected that production of ethanol will go back up in 2021. The ethanol co-products generated in 2020 included 29.4 million metric tons (MT) distillers grains, 3.1 million MT corn gluten feed, 0.6 million MT corn gluten meal, 1.5 million MT corn oil and 2.1 million MT captured $CO_2$. Corn was the major feedstock, which accounted for 93.2% of the total ethanol production in 2020.

Other feedstocks included sorghum, corn fiber, waste sugars, waste starch and lignocellulosic biomass. The interest in biomass as a potential feedstock for ethanol production continued to be strong. Despite previous failures by large corporations such as Poet and DuPont, New Energy Blue announced in July 2021 the plan to construct a full-scale ethanol biorefinery, which would consume 250,000 MT per year agricultural residues generated locally for ethanol production. The proposed plant would use the Inbicon process, which would also produce a lignin co-product for use as a solid biofuel and a natural binder and possibly xylitol as a specialty chemical with potential food applications [60,61].

### 3.1.2. Canada

Canada is a net importer of ethanol since the volume consumed in Canada generally exceeded the ethanol this country produced. In 1980, Canadian produced merely 8000 L but in 2010 the annual ethanol production of Canada reached 1.9 billion L. In 2019, ethanol consumption in Canada reached 3.33 billion L compared to 2 billion L in production. However, ethanol production in Canada has grown in recent years largely due to changes in feedstocks and increased capacity at existing ethanol facilities in Canada.

In 2020, Canada was ranked as the sixth largest ethanol producer in the world. The ethanol production in Canada represented 1.6% of the total global production. The two major feedstocks used for ethanol production were corn and wheat, which contributed 1534.3 million L and 360.7 million L, respectively [59]. Winter barley has been considered as a potential feedstock for ethanol production. However, this grain has not been used in any commercial ethanol production plant.

### *3.2. South America*

### 3.2.1. Brazil

Brazil continued to be the second largest ethanol producer in the world. In 2019, Brazil ethanol production including anhydrous and hydrous ethanol amounted to more than 35.3 million $M^3$, an increase of nearly seven percent in comparison to a year of 2018. This is the highest volume of fuel ethanol that Brazil has produced in the decade. In Brazil, sugarcane bagasse is commonly used as boiler fuel to produce energy to supply sugar mills. This is practiced to minimize the energy costs and also as an alternative to the utilization of the biomass left over.

In 2020, the ethanol production in Brazil represented 26.7% of the total global ethanol production [53]. There were 360 first-generation (sugar cane, corn) ethanol plants in Brazil in 2020 with a total nameplate capacity of 42,800 million L. However, only 67% of the capacity was used. There were three plants that used lignocellulosic feedstock with a total nameplate capacity of 127 million L. In 2020, these second-generation ethanol plants only operated at 25% capacity. The feedstocks included sugar cane (326.6 million MT), corn (6.0 million MT) and bagasse (178,000 MT). The total ethanol production in 2020 was 31.35 billion L, which included 32 million L produced from lignocellulosic feedstock (bagasse). The 2020 ethanol production was about 16% lower compared to the previous year (37.38 billion L). The decrease in ethanol production in 2020 was mainly caused by the diversion of sugar cane juice toward more sugar production in sugar-ethanol plants. The co-products included 120.1 million MT bagasse, 1.88 million MT DDGS and 108,000 MT corn oil. The traditional feedstock for first-generation ethanol production in Brazil is sugar cane. However, corn has become an important feedstock and its use for ethanol production has steadily gained ground. In 2020, 2.5 billion L ethanol was produced from corn, which was 1.17 million L higher than the previous year. The Corn Ethanol National Union (UNEM) predicted that corn ethanol production would reach 8 billion L by 2028. There currently are 11 corn ethanol plants, which include nine full-plant types (corn only) and 2 flex-plant types (corn and sugarcane). Two full-plants and one flex-plant are currently under construction [55].

### 3.2.2. Argentina

In 2020, Argentina was ranked 8th in the world with a contribution of 1.0% toward the total global ethanol production [53]. There were 22 ethanol plants with a total nameplate capacity of 1580 million L per year. The ethanol plants, however, were operated only at 55.1% capacity. Corn and molasses were used as feedstocks at 1.09 million MT and 1.70 million MT, respectively. The total ethanol production was 870 million liters, which was significantly lower than the total production in the previous year (1073 million L). The DDGS co-product was also produced at 345,000 MT [62].

### 3.2.3. Colombia

Colombia was ranked 13th in the world in 2020 with a contribution of 0.44% toward the total global ethanol production [60]. The country had six ethanol plants, which used sugar cane as the only feedstock. The total nameplate capacity of the plants was 540 million liters. These plants were run at only 73.1% capacity in 2020. The total ethanol production in 2020 was 395 million liters with 1.34 million MT bagasse co-product. Five of the six ethanol plants were directly linked to the sugar production plants. The bagasse obtained after juice extraction was used to generate energy for internal use in these plants. Most ethanol plants in Colombia were energy self-sufficient and even generated surplus energy to sell to the national grid [63].

### 3.3. Europe

#### The European Union

The total ethanol production in the EU in 2020 was 4.8% of the total global production [53]. The major feedstocks used for ethanol production were sugar beets (7.45 million MT), corn (6.35 million MT), and wheat (2.64 million MT). Other first-generation feedstocks were triticale (1.04 million MT), rye (520,000 MT) and barley (450,000 million MT). Wheat was predominantly used in Belgium, Germany, France and the UK. Corn was the preferred feedstock in Hungary, where corn was abundantly available, and in the Netherlands, Spain, and the UK, where the majority of corn came from the Ukraine to provide the feedstock for ethanol plants, which are located near seaports. The inland ethanol plants in Spain used a combination of corn and barley as the feedstock. Sugar beets and their derivatives were used for ethanol production in France, Germany, the UK, the Czech Republic, Belgium and Austria. The usage of lignocellulosic feedstocks in 2020 was doubled to 200,000 MT compared to the previous year. There were 57 first-generation ethanol plants with a total nameplate capacity of 8.15 billion L, which were operated at 58% capacity. There were also 3 lignocellulosic ethanol plants with a total nameplate capacity of 90 million L. However, these second-generation ethanol plants were operated at only 28% capacity. Five other lignocellulosic ethanol plants were under construction in Finland (sawdust, 10 million L per year), Italy (biomass, 28 million L per year), Austria (wood sugars, 30 million L per year), Romania (wheat straw, 65 million L per year) and Bulgaria (corn stover, 50 million L per year). All of these plants are expected to be in operation soon. The ethanol production in the EU in 2020 was 5.47 billion L. Due to the COVID-19 pandemic, ethanol consumption in the EU has decreased by 10.1%. However, this number was still slightly lower than the 13.0% decrease in gasoline consumption. According to the 2021 European Union Biofuels Annual Report, the co-products included 3.33 million MT DDGS and 188,000 MT corn oil [56].

### 3.4. Asia and Rest of the World

#### 3.4.1. Vietnam

Vietnam is an agriculture-based country and has abundant natural resources for renewable power development. However, the pace of renewable energy development including ethanol production is not yet rapid due to barriers such as the small size of the country's economy, the lack of financial capacity, advanced technologies and human

resources. Institutional barriers such as market-controlled mechanisms and unstable supporting policies also limit the development of renewable power sectors [64].

In terms of ethanol development, due to the economic development, Vietnam has consumed lots of gasoline in recent years. The United States Department of Agriculture (USDA) estimated gasoline consumption of Vietnam in recent years has grown approximately 4–5 percent per year. In the first quarter of 2020, Vietnam spent about $2.5 billion on importing crude oil and petroleum products, including ethanol, in addition to its production [65]. Since 2007 ethanol supporting regulations have gradually increased, however, the commercialization and sales of five percent ethanol blended gasoline (E5) was pushed back to 2018 due to a lack of understanding of the environmental benefits of ethanol blended gasoline by consumers, together with some persistent rumors and myths that usage of ethanol might harm vehicle engines.

### 3.4.2. Korea

Korea is the 7th ranked $CO_2$ emitter in the world and similar to other countries is concerned about its high $CO_2$ emissions and its dependence on imported crude oil. All of the oil Korea consumes is imported from foreign countries. South Korea is a main importer of United States ethanol, with a total volume of 263.84 million L (24.7 million bushels in corn equivalent) in 2017–2018. In the mid-1990s, production of ethanol for fuel use in Korea using imported cassava as a feedstock was initiated. The Korea Ministry of Knowledge Economy (KMKE) announced an action plan to increase the use of biofuels in the transport sector from 0.2 billion L in 2008 to 5 billion L by 2030 [66].

### 3.4.3. China

From 2004 to 2016, China produced ethanol with an average annual increase in production rate of 16.8 percent. In 2017, China had an ethanol production of 2.8 million MT. In 2018, China produced 6.6 million MT, making it the fourth-largest ethanol producing country/region in the world, after the United States, Brazil, and the EU. China set a policy that gasoline supplies across the country were required to be blended with ethanol by 2020. According to a news source, China aimed to have 15 million MT by 2020, which is almost triple its current ethanol production capacity, in order to keep up with growing demands for cleaner fuels [57]. The target would exceed the estimated domestic production capacity of China and the country would need to import ethanol from foreign countries such as the United States and Brazil. Corn is China's main feedstock, which is currently accounting for 64 percent of total output for China's ethanol production [67].

### 3.4.4. India

India ranked 6th among the leading ethanol producers in the world. In 2020, India still remained one of the biggest importers of the United States ethanol, with a market share of 99 percent. With Modi's "self-resilient" strategies, India sets its ambitious goal of E-20 by 2025 while retaining its immediate goal of E-10 by 2022. India has a total installed ethanol capacity of 5 billion L, of which molasses-based distilleries constitute 4.2 billion L, or 85 percent of the overall production capacity, while grain-based distilleries constitute 750 million L (equivalent to 15 percent).

Formerly, ethanol in India could only be produced from molasses or sugar juice. However, India's existing ethanol is produced from a variety of feedstocks such as cereals (rice, wheat, barley, maize, and sorghum). In 2020, an estimated 2.98 billion L of ethanol was produced from molasses. In 2021, India's ethanol production was forecast at 3.17 billion L, 7% above 2020 due to surplus sugarcane production, and 2021 average ethanol blending rate in gasoline of India was estimated at 7.5 percent, due to accelerated government efforts to divert more feedstock toward ethanol [68].

### 3.4.5. Thailand

Thailand is the 7th largest ethanol producer in the world. Thailand consumed 1500 million L of ethanol in 2020. Thailand's 20-year Alternative Energy Development Plan (AEDP) for 2018–2037 targeted 2.0 billion L for ethanol in 2021 and 2.7 billion L for ethanol consumption in 2037. Molasses is the primary feedstock for ethanol production in Thailand. Molasses supplies in Thailand have been tight due to reduced sugarcane production for two consecutive years. Ethanol demand is primarily expected to be fulfilled by cassava-based ethanol in Thailand [69].

### 3.4.6. Australia

Australia's ethanol consumption is forecast to remain stable in 2020 at only 1.4% of gasoline consumption. Ethanol in Australia is primarily produced as a biofuel for passenger and commercial vehicles and also for alcoholic beverage, industrial chemicals, and solvents used in pharmaceutical and cosmetic applications. In spite of large feedstock availability, Australia's ethanol production volumes remain small because there is no nationwide fuel ethanol program. In Australia only two states, which are New South Wales and Queensland, have mandates with a fuel ethanol program, and have the highest consumption of ethanol blended fuel.

In New South Wales, Manildra is the largest ethanol producer with a capacity of over 300 million L. In this plant wheat starch is processed through an integrated process which separates the gluten and processes the remaining starch into a range of food and industrial-grade starches, glucose syrups, and ethanol products. In Queensland, a plant at Dalby producing ethanol from starch-based feedstock is operated by United Petroleum. This biorefinery with a capacity of 80 million L fuel-grade ethanol is located in a sorghum growing region in the Darling Downs and processes up to 0.2 million MT of sorghum grain a year from local growers. At full capacity, the biorefinery can also produce 830,000 MT of wet distillers grain, which is used for animal feed supplements [70].

### 3.4.7. Zimbabwe and Other African countries

Zimbabwe has adopted ethanol petrol blending regulations. Two ethanol plants in Zimbabwe, namely Triangle Sugar and Hippo Valley, were installed. In 2010, the Triangle plant resumed ethanol production after refurbishing its ethanol plant with a capacity of 27 million L per year. The government of Zimbabwe also initiated the Chisumbanje sugar/bioethanol project with an aim of using 10,000 hectares of sugarcane for this project. In March 2012, owners of this project halted production because the plant was running out of storage space. In Zimbabwe, some oil companies were only selling E10 at a few of their filling stations [71].

In other African countries, lead additives are still heavily used in gasoline and where sugarcane production cost is high, ethanol can be a cheap source of octane in gasoline. It is estimated that in Africa, to replace all the lead used in African gasoline, this would require Africa to produce about 20% of the amount of ethanol currently produced in Brazil, and would require the shift of some sugar production to ethanol production. African countries that could replace lead with ethanol using primarily their by-product molasses production include Zimbabwe, Kenya, Egypt, Zaire, Zambia, Sudan, Swaziland, and Mauritius at a more modest scale [72].

## 4. Discussion

Over the last several decades, ethanol has emerged to become an important transportation liquid fuel. The corn ethanol in the United States and the sugarcane ethanol in Brazil have gained their position as a mature industry. The strong interest in ethanol from sustainable sources has led to the development of process technology for production of ethanol from biomass. The research in this area has led to many technological achievements and changes of the biobased product industry. One example is the development of biomass pretreatment processes, which have been improved to shift the focus on the

cellulose fraction in biomass to include hemicellulose utilization. In the early developments of pretreatment processes, lignin was considered as a waste, which was to be burned to provide the thermal energy required in a biomass ethanol plant. The research on processes for conversion of lignin to high-value products has changed this view and lignin now is considered as a co-product of biomass ethanol production. Another example is the developments of much more efficient of commercial cellulase and xylanase products. Despite several failures of biomass ethanol production plants, others continued to be constructed with improved technical knowledge and better production strategies. It is anticipated that the second-generation ethanol industry will eventually emerge and establish itself similar to the first-generation ethanol based on starch-based and sugar-based feedstocks.

## 5. Conclusions and Global Trend Forecast

As discussed previously, corn ethanol in the United States and sugarcane ethanol in Brazil have been in commercial practice for many years and hence there is little room for research to significantly improve the technology. However, in the area of biomass ethanol, there are still opportunities for research to improve the technology and to bring it closer to commercial success. One area that needs further research is the development of high-value co-products that can be produced from the C5 sugars and the processes for their production. Examples of these products are xylitol and astaxanthin, among others. Another area that still offers many research opportunities is lignin utilization. Similar to the case of the C5 sugars, lignin can be used as a feedstock for production of high-value coproducts and development of processes for their production still requires significant research efforts.

COVID-19 has hit the world and the production industries worldwide have been seriously affected. As a result, ethanol output worldwide dropped in 2020. Other areas are being hit as well but their lower reliance on the fuel ethanol markets make the negative impact of the pandemic have been relatively less. When the pandemic is over and the production industries return to normalcy, it is expected that the rebound will come, however, the production might not be as strong as before the lock-down period worldwide. During the pandemic, even after the restrictions are eased, demand on ethanol for hand-sanitizer manufacturing is high. In developing countries, stronger emissions regulations and policies help to encourage the production potential for biofuels to offset $CO_2$ emissions. With an eye toward the future, worldwide ethanol production is expected to increase again.

**Author Contributions:** Conceptualization, T.-D.H. and N.N.; methodology, T.-D.H. and N.N.; validation, T.-D.H. and N.N.; formal analysis, T.-D.H. and N.N.; investigation, T.-D.H. and N.N.; resources, T.-D.H. and N.N.; data curation, T.-D.H. and N.N.; writing—original draft preparation, T.-D.H. and N.N.; writing—review and editing, T.-D.H. and N.N.; visualization, T.-D.H. and N.N.; project administration, T.-D.H. and N.N. All authors have read and agreed to the published version of the manuscript.

**Funding:** This research received no external funding.

**Institutional Review Board Statement:** Not applicable.

**Informed Consent Statement:** Not applicable.

**Data Availability Statement:** Not applicable.

**Conflicts of Interest:** The authors declare no conflict of interest.

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
