# Peer review of "Recent Developments and Current Status of Commercial Production of Fuel Ethanol"

_fermentation, doi:10.3390/fermentation7040314_

Round 1
Reviewer 1 Report
Overall english and scientific writing of MS is good. The topic selection of review article is also quite impressive particularly the analysis of ethanol production levels in different countries. Review can be accepted after revision.

Reviewer 2 Report
The manuscript from Hoang and Nghiem describes the technological and economic developments and trends in the Ethanol production. The review describes very well the global directions in the field.
The manuscript is well written. However, the review has minor points.
The introduction should go deeper into the developments of the field for the production of this important biofuel and what the differences in terms of sustainability for the production of bioethanol are present compared to fossil-based fuels.
The manuscript contains very few citations compared to the vast field that the authors are describing. The reviewer suggests to add more citations for each section. Especially, for the section of lignocellulosic biomass conversion and to gasification, the reviewer suggests to add the following review:
Ciliberti et al Processes 2020 "Syngas Derived From Lignocellulosic Biomass Gasification as an alternative resource for Innovative Bioprocesses.
The reviewer also suggests to add figures throughout the manuscript that can describe better the processes described in the text. This could improve the interest from readers and the understanding of each process.
In raw 225, the reviewer talk about a process called pretreatment. However, there are many pretreatment processes that can be used for the release of the polysaccharides used as raw materials in hydrolysis for sugar release. The reviewer suggests to rephrase this part for better clarity.
The authors describe that S. cerevisiae is able to metabolize sugars such as sucrose, glucose and fructose. Then, in raw 233, the authors describe that the yeast can metabolize only glucose, which is not correct. The reviewer suggests to rephrase this part focusing on C6 and C5 sugars. The latter class contains also xylose which is not metabolized by wild-type strains and, as the authors suggest, modifications can improve this metabolic pathway.
Reviewer 3 Report
Major Revision:
This paper reviews recent developments and status of commercial production of ethanol across the world. The study also examines current technologies in ethanol production processes used for each type of feedstock, both currently practiced at commercial scale and newly developed technologies, and production trends in various regions and countries in the world.
The paper meets the aim and the scope, as well as, the high academic standards of the ‘Fermentation’ Journal. However, the following specific improvements should be made, before accepting the paper for possible publication to the Journal.
General comments:
- Please do not split the words at the end of the line. Please revise this grammar error throughout the manuscript. For example in lines 2, 5, 10, 11, 12, 17, 18, 20, 21 etc.
Abstract:
- Line 13: I suggest you provide the latest data on global ethanol production in 2020 instead of 2019.
- Please include the knowledge gap and the major conclusions of the paper.
- Introduction
- Line 29: Please be more specific with the term “Source” or give a short description.
- Since the idea of reviewing the recent developments and status of commercial production of ethanol, is not a new one, at the end of this section, the authors should provide a clear and concise understanding of the primary contribution of their manuscript.
- Production technologies
- An overview table of ethanol yields for each type of feedstocks is strongly recommended to be added, including also a SWOT analysis with a critical view of each feedstock processing.
- Simplified process schemes in each feedstock fermentation process are strongly recommended to be added.
- Line 229-230: Please include the major conclusions of the biomass pretreatment processes review.
- Global ethanol production
- A comparative table of the investigated countries, including the capacity of ethanol plants, consumption, total ethanol import, the main utilization of ethanol consumption (i.e. energy generation, transport sector, etc.), and key aspects (i.e. regulations goals, perspectives, opportunities, etc.) is strongly recommended to be added. For comparison purposes, all figures should be given in the same units.
- A Discussion part should be included in order to describe in detail the findings of the study. You could include table 1 and discuss in a broadest context possible the main findings of the table. Please write this section according to the template provided by Fermentation Journal in the following link: https://www.mdpi.com/journal/fermentation/instructions
- Conclusions
- Conclusion must be contain basic findings of the review also with the knowledge gap the paper would be filling. Additionally, this section is poor; more discussion is required.
- Kindly provide strong recommendations for future researches.
References
- The number of references is quite low. Similar reviews have over 80 references. The context of this paper should be enriched with studies published in the Fermentation Journal.
- The reference list does not comply with instructions for authors. Please revise it accordingly.
Round 2
Reviewer 3 Report
Recommendation: Authors Should Prepare a Minor Revision
In the revised manuscript the authors have satisfactorily addressed most of the issues I had raised in my original review. The quality of the article has been significantly improved. However, the paper still needs a minor revision before being accepted for possible publication to the Journal.
Specifically, in the revised manuscript, the authors have provided comparative tables 3 and 4 of the investigated countries in the Section 3.4.7. I claim that I would have liked this information to be given in a separated Section or in the Discussion part.
